# Recent Research in Ocular Cystinosis: Drug Delivery Systems, Cysteamine Detection Methods and Future Perspectives

**DOI:** 10.3390/pharmaceutics12121177

**Published:** 2020-12-03

**Authors:** Ana Castro-Balado, Cristina Mondelo-García, Iria Varela-Rey, Beatriz Moreda-Vizcaíno, Jesús F. Sierra-Sánchez, María Teresa Rodríguez-Ares, Gonzalo Hermelo-Vidal, Irene Zarra-Ferro, Miguel González-Barcia, Eva Yebra-Pimentel, María Jesús Giráldez-Fernández, Francisco J. Otero-Espinar, Anxo Fernández-Ferreiro

**Affiliations:** 1Pharmacy Department, University Clinical Hospital of Santiago de Compostela (SERGAS), 15706 Santiago de Compostela, Spain; ana.castro.balado@gmail.com (A.C.-B.); crismondelo1@gmail.com (C.M.-G.); iria.varela.rey@sergas.es (I.V.-R.); irene.zarra.ferro@sergas.es (I.Z.-F.); miguel.gonzalez.barcia@sergas.es (M.G.-B.); 2Pharmacology Group, Health Research Institute of Santiago de Compostela (FIDIS), 15706 Santiago de Compostela, Spain; zalohermelo@gmail.com; 3Department of Pharmacology, Pharmacy and Pharmaceutical Technology, Faculty of Pharmacy, University of Santiago de Compostela (USC), 15782 Santiago de Compostela, Spain; beatriz.moreda@rai.usc.es; 4Pharmacy Department, Hospital de Jerez de la Frontera, Jerez de la Frontera, 11407 Cádiz, Spain; quequesierra@hotmail.com; 5Ophthalmology Department, University Clinical Hospital of Santiago de Compostela (SERGAS), 15706 Santiago de Compostela, Spain; trares2@gmail.com; 6Department of Applied Physics, Optometry, Faculty of Optics and Optometry, University of Santiago de Compostela (USC), 15782 Santiago de Compostela, Spain; eva.yebra.pimentel@usc.es (E.Y.-P.); mjesus.giraldez@usc.es (M.J.G.-F.)

**Keywords:** cystinosis, ophthalmic administration, cysteamine, drug delivery systems, analytical chemistry methods

## Abstract

Cystinosis is a rare genetic disorder characterized by the accumulation of cystine crystals in different tissues and organs. Although renal damage prevails during initial stages, the deposition of cystine crystals in the cornea causes severe ocular manifestations. At present, cysteamine is the only topical effective treatment for ocular cystinosis. The lack of investment by the pharmaceutical industry, together with the limited stability of cysteamine, make it available only as two marketed presentations (Cystaran^®^ and Cystadrops^®^) and as compounding formulations prepared in pharmacy departments. Even so, new drug delivery systems (DDSs) need to be developed, allowing more comfortable dosage schedules that favor patient adherence. In the last decades, different research groups have focused on the development of hydrogels, nanowafers and contact lenses, allowing a sustained cysteamine release. In parallel, different determination methods and strategies to increase the stability of the formulations have also been developed. This comprehensive review aims to compile all the challenges and advances related to new cysteamine DDSs, analytical determination methods, and possible future therapeutic alternatives for treating cystinosis.

## 1. Introduction

Cystinosis is a rare autosomal recessive disease that affects about 1 in 100,000–200,000 people among the general population [1], characterized by high levels of cystine within the lysosomes in cells of certain types of tissues. The accumulation of this substance is caused by mutations in the CTNS gene which codes for cystinosin, the carrier that transports cystine out of the lysosome [2,3]. The presence of cystine crystals in different tissues leads to the progressive impairment and dysfunction of multiple organs, such as kidneys, pancreas, brain, thyroid and eyes [4]. The disease, if it remains untreated, results in death from renal failure by the second decade of life [5].

Although renal damage prevails in premature forms of the disease, all forms of cystinosis affect ocular structures: cornea, conjunctiva, iris, ciliary body, choroid, retina and optic nerve. The most frequently described ocular manifestation is the deposition of cystine crystals in the cornea, but the exact mechanisms of crystal formation are not yet fully understood. Stromal crystals have a needle shape and are oriented parallel to the corneal stromal lamellae [6], which suggests that the structure of collagen in the stroma plays a very important role in the evolution of cystine crystals [7]. This deposition is one of the most troublesome complications affecting the quality of life of patients with cystinosis, especially as the prognosis improves and life expectancy increases, causing photophobia, visual impairment and, finally, blindness [8,9]. In addition, this accumulation with time can cause corneal scars, keratitis and cataracts [10].

Corneal crystals are visible on ophthalmological examination as of 16 months of age in most patients [11]. Recent exploration techniques with in vivo confocal microscopy and optical coherence tomography (OCT), anterior segment optical coherence tomography (AS-OCT) and in vivo confocal microscopy (IVCM) of the anterior segment, are useful methods to evaluate crystals and detect their morphological characteristics and corneal alterations [12,13].

Early diagnosis and treatment of the ocular manifestations of childhood nephropathic cystinosis are essential. Therefore, early initiation and adherence to topical therapy have a significant impact on disease progression [8]. Nowadays, although new prodrug strategies based on *N*-acyl or glutaric acid derivatives of cystamine have been researched [14,15], the aminothiol cysteamine remains the only available treatment of cystinosis [16]. It lowers intracellular levels of cystine by forming a cysteamine–cysteine mixed disulfide, which can egress the lysosome using the undamaged excretion pathway for lysine [17] (Figure 1). The authorization of oral cysteamine (Cystagon^®^) by the Food and Drug Administration (FDA) in 1994, and by the European Medicine Agency (EMA) in 1997, has totally changed the management and prognosis of the patients with cystinosis [11,18,19]. However, it does not prevent corneal crystal accumulation because it cannot reach the cornea due to its lack of vascularization [20]. Consequently, it is necessary to instill cysteamine on the ocular surface in order to eliminate the cystine crystals at this location [11].

In 2012 Cystaran^®^ (0.44% cysteamine ophthalmic solution) was approved by the FDA as an orphan drug indicated for the treatment of corneal cystine crystal deposits. The posology indicated in the prescribing information is one drop in each eye, every waking hour [21]. This posology, with its very frequent administrations, complicates patient’s adherence to the treatment. In order to improve this aspect, in 2017 Cystadrops^®^ (0.55% cysteamine ophthalmic solution) was approved by the EMA and by the FDA in 2020 [22]. It contains sodium carmellose which provides a high viscosity to the formulation, achieving a longer residence time on the ocular surface and allowing a dosage of just four times per day [23]. Therapeutic options marketed for ophthalmic treatment are scarce. Although ethical issues sometimes should outweigh economical and feasibility issues, the pharmaceutical industry does not allocate sufficient resources for the study of rare diseases and the development of orphan drugs. Because the commercialized cysteamine presentations are not available in most countries, hospital pharmacy departments are responsible for preparing homemade eyedrops as a therapeutic alternative [24]. The problem is that on many occasions, these formulations lack exhaustive stability controls under different storage conditions, cysteamine being a very easily oxidizable molecule [25].

Concerning topical treatments for ocular cystinosis, the patients’ therapeutic compliance is a major factor. In this regard, a new topical treatment, which only needs administrations every several hours or even days, would improve it [5]. The problem of the necessary frequent administration of topical cysteamine stems from the fact that eye drops are rapidly cleared from the ocular surface due to reflex tearing, constant blinking and nasolacrimal drainage resulting in a short contact time on the eye [26]. In addition, this organ has the tendency to maintain its residence volume at approximately 10 µL and, consequently, the bioavailability of a topically applied drug is typically <5% [27].

In this context, the development of new drug delivery systems (DDSs) which can overcome the high ocular clearance of conventional cysteamine eye drops is a major issue in ophthalmology research [28]. The increase of patient compliance with systems that delay the delivery of cysteamine and, consequently, minimize its frequency of administration is a key factor for improving the treatment of the pathology [20]. The aim of this comprehensive review is to make an overview of the new challenges in ocular cystinosis topical treatment, including cysteamine determination methods, developed DDSs and possible future therapeutic alternatives.

## 2. Drug Delivery Systems (DDSs)

Ocular drug delivery has always been a challenge due to the limitations of conventional eye drops. The high tear turnover rate (1 µL/mL), loss of drug due to rapid blinking, reflex tear production, the tear-film barrier and high nasolacrimal drainage are factors that limit the absorption of topically applied ophthalmic formulations. To increase the drug bioavailability, a higher residence time on the ocular surface needs to be achieved. During the last decades, research has been done on the development of DDSs offering longer retention and a sustained release of the drug molecule to pass through these barriers [29]. The variability of DDSs intended for ophthalmic administration has experienced a large increase in the last decade. However, the difficulties associated with the stability of cysteamine, combined with the fact that cystinosis is a rare disease, means the development of DDSs intended to treat this disease has been more limited than in other ophthalmic diseases (Figure 2).

### 2.1. Hydrogels

Hydrogels are networks of polymer chains extensively swollen with water, which they retain within their structure. Their porosity and high-water content make hydrogels suitable for encapsulation of water-soluble drugs, since they are processed at room temperature and organic solvents are rarely needed [30,31].

Hydrogels can be designed from natural or synthetic polymers. Natural polymers present the advantage of minimal toxicity due to their high biocompatibility. However, their main drawback is their considerably shorter drug release compared to synthetic hydrogels, which limits their use as long-term sustained DDS [31,32].

By formulating cysteamine as a bioadhesive ophthalmic gel with controlled drug release, the administration frequency could be reduced and, consequently, increase therapeutic compliance. Particularly, hydrogels which are transparent and bioadhesive are highly desirable for topical ophthalmic application [10]. Concerning the treatment of ocular cystinosis, several types of polymers have been used for the development of topical hydrogels.

#### 2.1.1. Synthetic Hydrogels

Synthetic hydrogels based on poly(acrylic acid) (PAA), commercially available as Carbopol^®^, can be obtained crosslinked with allylsucrose or allylpentaerythrol (carbomer) for pharmaceutical application. PAA is highly coiled and tightly packed but once dispersed in water the polymer swells to form a colloidal dispersion that behaves as an anionic electrolyte. In addition, the nonNewtonian pseudoplastic rheology of PAA hydrogels enhances the process of blinking because it causes an important reduction in apparent viscosity as a function of the high external shear-stresses applied by the eyelid [5]. Considering these properties, Buchan et al. designed a hydrogel composed of carbomer 934 for the topical administration of cysteamine on the ocular surface. The formed hydrogel was bioadhesive, transparent and offered significantly less resistance to blinking than Newtonian liquids of equivalent consistency, resulting in longer contact times on the surface of the eye. Furthermore, dissolution studies showed a first-order release of the active drug from the sample matrix with no destruction of the gel properties due to the addition of cysteamine. Accordingly, the authors affirm that this kind of hydrogels based on pseudoplastic fluids form weak networks with desirable properties to increase the residence time of cysteamine on the ocular surface [5].

For their part, McKenzie et al. carried out rheology, bioadhesion, dissolution and stability studies with several synthetic polymers in order to test their suitability for ophthalmic delivery. In this sense, they agree with Buchan et al. regarding the fact that carbomer 934 is suitable for ophthalmic delivery of cysteamine. However, their studies showed problems related to gel opacity [10].

Although both previous formulations have been characterized, they lack in vivo studies which are indispensable to show the permanence of the gels on the ocular surface, especially considering that the aim of the studies was to extend the retention time of the cysteamine.

#### 2.1.2. Natural Hydrogels

Regarding ophthalmic delivery of cysteamine using natural hydrogels, Bozda et al. synthetized viscous solutions of cysteamine hydrochloride by using hydroxypropylmethyl-cellulose (HPMC) and evaluated in vitro characteristics and stability. All the viscous solutions tested showed nonNewtonian flow behavior. Concerning in vitro release tests, they revealed that more than 80% of cysteamine hydrochloride was released from the HPMC solutions in 8 h. In addition, the formulations produced no irritation when they were tested on rabbit eyes [33].

On the other hand, McKenzie et al. showed that sodium hyaluronate and hydroxyethyl cellulose were both suitable for ophthalmic delivery of cysteamine. Among the obtained results, it is necessary to highlight the fact that sodium hyaluronate displayed optimum performance in the preformulation tests, being pseudoplastic and bioadhesive, as well as releasing cysteamine over 40 min [10].

In these sense, Luaces-Rodríguez et al. [9,24] selected two different polysaccharide hydrogels to formulate cysteamine: an ion sensitive hydrogel with the polymers gellan gum and kappa-carrageenan, and another composed of hyaluronic acid [10,34]. In this regard, the authors performed in vitro (characterization of the hydrogels, drug release and cell toxicity) and ex vivo (transcorneal permeation and Hen´s Egg Chorioallantoic membrane (HET-CAM)) assays. On the one hand, in vitro release studies determined that both hydrogels can control the release of cysteamine over time, showing zero-order kinetics for 4 h. At the same time, they affirmed that these hydrogels could act as corneal absorption promoters, as they allow a higher permeation of cysteamine through bovine cornea compared to a solution. On the other hand, ex vivo assays showed no irritation on the ocular surface. Finally, the authors also accomplished in vivo studies based on direct measures of biopermanence time by positron emission tomography (PET), demonstrating that both formulations presented a high retention time on the ocular surface of rats [9]. Hydrogels previously described, as well as their main characteristics, are listed in Table 1.

In spite of the fact that important advances have been made regarding the development of new hydrogels, further studies should be carried out. Specifically regarding the development of smart hydrogels which can be very advantageous because they dramatically change their volume and other properties in response to environmental stimuli such as temperature or pH [35]. In addition, the synthesis of hydrogels based on nanoparticles could be a very promising strategy, which has provided interesting results in other pathologies [36].

### 2.2. Nanowafers

Nanowafers are tiny transparent circular discs that can be applied on the ocular surface with a fingertip [37]. Marcano et al. synthetised cysteamine-nanowafers via a hydrogel template strategy. In this study, they fabricated poly(vinyl alcohol) (PVA) nanowafers loaded with cysteamine. They contained arrays of drug-loaded nanoreservoirs from which the drug was released in a tightly controlled way for an extended period of time. At the end of this period, the nanowafer dissolved and faded away. Cysteamine-nanowafers are highly transparent; in fact, the refractive index of a cysteamine-nanowafer is very close to that of a soft contact lens. Hence, nanowafer application on the cornea does not affect the normal vision. These authors found that cysteamine was stable in the nanowafer and in a therapeutically effective form for up four months when stored at room temperature [26].

In addition, they carried out in vivo studies to determine the efficacy of the cysteamine-nanowafer in comparison to topical cysteamine eye drop formulation (0.44%). Two groups of CTNS−/− mice (three per group) were treated with the cysteamine-nanowafer (10 μg of cysteamine, once a day) and cysteamine eye drops (5 μL, 22 μg) twice a day for 30 days. These studies revealed that compared to the baseline corneal cystine crystal volume, cysteamine eye drops reduced the crystal volume by 55%, while cysteamine-nanowafer reduced the crystal volume by 90%, confirming that the cysteamine nanowafer treatment was significantly more efficacious. The authors stated that this higher efficacy was due to the longer residence time of the drug molecules on the eye achieved with the nanowafer, which enabled their diffusion into the ocular surface epithelium [26].

The development of this nanowafer means a very innovative and interesting alternative to properly control cysteamine release on the ocular surface. In this sense, other studies are needed to explore the possibilities that nanosystems, which have demonstrated successful results for other pathologies, could offer for ocular cystinosis treatment [36,38].

### 2.3. Contact Lenses

#### 2.3.1. Contact Lenses as Drug Delivery Systems

Contact lenses (CLs) are medical devices widely used by over 125 million individuals in the world to correct vision problems [39], and were proposed as ocular DDSs since the first prototypes were synthesized nearly 50 years ago. CLs are polymeric structures formed after a polymerization process of different monomers in the presence of a crosslinking agent. This system absorbs a large amount of water (30–80%) to form hydrogels having an aqueous phase permeable to oxygen. Depending on the nature and proportion of the different monomers that make up the polymeric structure of the CLs, conventional hydrogels or silicone hydrogels can be obtained. Modern materials currently used are an evolution of the well-known lens materials based on poly-2-hydroxyethylmetacrilate (p-HEMA) and silicone hydrogels [40,41].

Over the last two decades, the use of CLs as DDSs has been studied for the treatment of numerous ophthalmic pathologies [42]. Once a CL is placed over the eye, a thin layer of fluid is formed between the lens and the cornea, which takes about 30 min to dilute. When a drug is included in this layer, the time of contact between the drug and the cornea would be increased and, therefore, bioavailability would increase to approximately 50% when compared to the administration of ocular drops (between 1–5%). Further, conjunctival absorption would diminish and, consequently, a smaller amount of drug would enter the systemic circulation, avoiding the appearance of adverse effects [39]. This increase in bioavailability would allow reduction of the dosage, enabling high therapeutic compliance for patients, and correct vision problems at the same time. The main limitation they present as DDSs is that polymers that make up the lenses and drugs have low affinity, leading to insufficient drug loading and too rapid delivery [43]. The main objective of many investigations in recent years has been to achieve a controlled and sustained drug release from the CL to the tear film located between the CL and the cornea so that corneal absorption can take place from there.

#### 2.3.2. Modified Contact Lenses

The simplest method for incorporating drugs into CLs is by immersing them in concentrated solutions of the active ingredient although, as previously mentioned, this technique tends to lead to insufficient charges and excessively rapid releases. To achieve controlled sustained release, a series of modifications have been studied, such as the incorporation of polymeric nanoparticles, microemulsions, micelles, liposomes, diffusion barriers (e.g., vitamin E) and sophisticated loading techniques such as molecular imprinting, ion ligand polymeric systems, drug-loaded films or supercritical fluid technology. With all that, it is important to note that the final modified lenses must maintain the conditions of oxygen permeability, transparency, comfort, water content, mechanical properties, ionic permeability, relatively neutral pH, tonicity and stability, preferably at room temperature [40,43,44].

Cysteamine is a hydrophilic small molecule (77.15 g/mol) and easily oxidizable, that shows little affinity for unmodified commercial CLs, giving rise to short releases and risking toxicity effects. Vitamin E can be incorporated into commercial silicone hydrogel CLs as a diffusion barrier, a strategy initially proposed by Peng et al. [45,46,47,48,49], achieving a higher bioavailability than eye drops. Vitamin E is a biocompatible hydrophobic molecule that exhibits low solubility in water, and which creates a tortuous pathway prolonging the drug diffusion time through the CL. Hydrophilic drugs like cysteamine must overcome the said obstacle to reach the ocular surface, increasing diffusion time. A vitamin E barrier is included through immersion of CLs in a solution of vitamin E in ethanol. The composition of the solution may differ according to the desired amount to be administered. Subsequently, the CL has to be placed in water to remove any remaining of ethanol, while the vitamin gets trapped within the CL. The antioxidant properties of vitamin E provide a certain degree of protection for a molecule as sensitive to oxidation as cysteamine. In addition, Vitamin E incorporation can block UV radiation, an additional benefit to cystinosis patients. On the other hand, even though the transparency of the CL is not affected, it can affect its size, oxygen diffusion and ionic permeability, which depend on the thickness of the barrier [39,50].

Hsu et al. studied the use vitamin-E-modified silicone-hydrogel CLs to extend the delivery of cysteamine. ACUVUE OASYS^®^ (Senofilcon A) with 19.14% vitamin and 1-DAY ACUVUE^®^ TruEye™ (Narafilcon B) with 10.22% vitamin E prolonged release durations about 3 h and 25 min, respectively. The mass of cysteamine released from the CLs decreased when vitamin E was incorporated. ACUVUE^®^ OASYS^®^ delivered about 583.4 μg of drug, but with the incorporation of 19.14% vitamin E this decreased to 408.8 μg. In the same way, the 1-DAY ACUVUE^®^ TruEye™ released 654.9 μg, but the mass released was reduced to 600.1 and 527.7 μg for 10.22% and 22.24% vitamin E loadings, respectively. Authors predicted that either a single 19.14% vitamin E loaded ACUVUE^®^ OASYS^®^ or two 10.22% vitamin E loaded 1-DAY ACUVUE^®^ TruEye™ per day were the safest options to deliver therapeutic doses of cysteamine to cornea. Results were based on an in vivo mathematical model used as an initial predictor of dose-related toxicity in animal studies. In this work it was also concluded that it was not clear whether a two hour release duration may be enough, since the current therapy uses 8–10 eye drops administered throughout the day [50].

Years later, Dixon et al. showed that vitamin E incorporation into CLs increases the duration of cysteamine release in silicone-hydrogels ACUVUE^®^ OASYS^®^ (Senofilcon A) and ACUVUE^®^ TruEye™ (Narafilcon A), the effect being more pronounced in TruEye™, possibly due to the low solubility of vitamin E in the lens matrix and higher aspect ratio of the barriers. The change observed in the composition of ACUVUE^®^ TruEye™ in regard to Hsu’s work, was due to the fact that since 2010, the brand gradually replaced Narafilcon B to Narafilcon A [51]. They also explored the effect of vitamin E incorporation in p-HEMA hydrogel CLs, ACUVUE^®^ Moist^®^ (Etafilcon A). Release durations were significantly increased in ACUVUE^®^ TruEye™ to 0.90, 2.0, and 4.0 h for vitamin E loadings of 10%, 20%, and 30%, respectively. For ACUVUE^®^OASYS^®^ lenses, release durations increased to 0.89, 2.15, and 4.25 h for 10, 20 and 30% loadings, respectively. Vitamin E incorporation did not significantly increase the release duration for hydrogel ACUVUE^®^ Moist^®^ lenses, suggesting that vitamin E did not form barriers in hydrogel lenses. This data, along with the high aspect ratio in silicone hydrogels suggests that barriers could be forming at the interface of the silicone and hydrogel phases. Both cysteamine mass loaded and partition coefficients decreased for increasing vitamin E loadings, as seen in Table 2. CLs would need to be worn for about 4–5 h each day, less time than the typical duration of daily-wear. Finally, they also showed that daily use of cysteamine-loaded CLs did not cause any adverse response in the eyes of rabbits over a seven-day use [52].

Recently, the same research group developed CLs with carbon black to obtain tinted lenses to mitigate photophobia reducing transmittance. The presence of cystine crystals in the cornea causes photophobia, or extreme light sensitivity, in patients with cystinosis [53] which, after years of squinting, can cause intractable blepharospasm [54,55]. This incorporation can be used in cysteamine-vitamin E-loaded lenses, maintaining controlled release and keeping the lens parameters when carbon black concentrations of 0.3% are used [56]. Despite the advances made in the development of CLs as cysteamine delivery systems, more studies are needed to allow for their optimization. For this, it is important to select those CLs that have a more suitable composition for correct loading and release, and to develop methods that allow better control of release without affecting any of the CL’s parameters and preserving the stability of the cysteamine. Composition, total amount of cysteamine released and release time of the previously described CLs are summarized in Table 2.

CLs are presented as DDSs easy to administer and comfortable to the patient with dosing schedules that lead to better adherence than those obtained under frequent instillation of ophthalmic drops. Despite this, CLs are not available on the market due to issues like drug stability during processing/manufacturing, achievement of zero-order release kinetics, avoidance of drug release during the post-manufacturing monomer extraction step, protein adherence, drug release during storage and cost-benefits. However, they have potential applicability in the field of ophthalmic compounding, as drugs and excipients can be easily loaded onto them under sterile conditions as part of a relatively simple low-cost formulation process [43].

## 3. Stability and Analytical Determination Methods

### 3.1. Cysteamine Structure and Stability Properties

Cysteamine (also known as β-mercaptoethylamine, 2-aminoethanethiol, 2-mercaptoethylamine, thioethanolamine and mercaptamine) is an aminothiol compound (HS-CH_2_-CH_2_-NH_2_) whose endogenous production occurs during the degradation of coenzyme A, when pantetheine is formed during the co-enzyme A metabolism cycle [57,58]. The efficacy of cysteamine in cystinosis treatment may be limited by its unpleasant organoleptic properties, strong hygroscopicity and chemical instability. The latter is especially important to take into account for the development of new ophthalmic formulations, being important to have a good determination method that allows its qualitative and quantitative detection, as well as distinguishing between its main oxidation metabolite cystamine.

Cysteamine exposure to air once the eye drop bottle is opened results in a short shelf life [25]. This degradation is due to a thiol functional group which immediately reacts with oxygen to produce a disulphide called cystamine, which is not effective at treating the cystine crystals located in the cornea [2,59]. This oxidation process takes place easily and rapidly in air and solution as a zero-order reaction, which indicates that the concentration of cysteamine decreases linearly with time [60]. The removal of oxygen from a solution of cysteamine by packing under nitrogen, and the addition of ascorbic acid, increase its stability. However, this technique showed lower efficacy in comparison with chelating agents such as disodium edetate [61,62]. In addition, it is also important to take into account the nature of the container, being preferable to use those materials that prevent the passage of oxygen inside. Other strategies to increase its stability have been used, such as the reduction of the pH (between 4.1 and 4.5) and freezing [63].

Dixon et al. evaluated two approaches for preventing the oxidation of cysteamine to increase the time of use after opening the bottle to one month. Firstly, they studied the use of antioxidants such as catalase enzyme and vitamins C and E, showing that catalase is the most effective additive, decreasing the oxidation rate by 58%. Secondly, the incorporation of diffusion barriers to prevent oxygen from reaching the cysteamine solution were investigated. This was accomplished by two methods: formulation of a hydrophobic layer which floats on the surface of the aqueous solution, and integration of OMAC^®^ oxygen resistant material into the eye drop bottle. Both delayed the onset of cysteamine degradation and decreased the rate of degradation [63].

In this sense, another important barrier for the development of cysteamine topical formulations is the fact that its physicochemical properties combined with its low molecular weight (77.15 g/mol) make its quantification a difficult issue, which is still trying to be solved nowadays [64].

### 3.2. Cysteamine Analytical Determination Methods

Diverse analytical methods have been proposed in the literature to detect and quantify cysteamine in nonbiological and biological samples (plasma and urine) such as high-performance liquid chromatography (HPLC) with fluorescence and ultraviolet (UV) detection, high-voltage electrophoresis, ion-exchange column chromatography, enzymatic assay and gas chromatography with flame ionization and photometric detection. The choice of the most adequate method for each situation is difficult because it must be based on several parameters such as the type of sample [60]. Cysteamine structure does not present a chromophore in its structure, so that conventional analytical methods with UV absorbance or fluorescence detection are not useful if the molecule remains unchanged. For its detection and quantification with these techniques, a derivatization process is necessary. In the last four decades, numerous derivatization agents were applied and optimized [65,66,67,68,69,70,71,72,73,74,75,76,77,78,79,80,81]. Most of the methods described below have not been specifically used for the determination of cysteamine from DDSs, but rather most of them have been purposed for the determination of cysteamine in biological samples and in cosmetic products.

#### 3.2.1. High-Performance Liquid Chromatography (HPLC)

Cysteamine detection by HPLC is performed using fluorescence, UV, and electrochemical detections. As mentioned before, when fluorescence and UV detectors are used, derivatization is usually needed [65,66,67,68,69,70,71,72,73,74,75,76,77,78,79,80,81], while for electrochemical methods this is not required. Derivatization agents, conditions and elution times are summarized in Table 3. Among methods depicted in this table, it is worth highlighting the one recently published by Kim et al. It is a simple and effective analytical method for the quantitation of cysteamine using reversed-phase HPLC for simultaneous determination of cysteamine and cystamine, in which the previous derivatization process is not necessary [64].

#### 3.2.2. Ion-Exchange Column Chromatography

The detection of cysteamine by ion-exchange column chromatography using an amino acid analyzer was accomplished in 1978 by Hsiung et al. Unfortunately, cysteamine was detected after 268 min, which is a long time that may lead to the oxidation of the product during the analysis [83]. Years later, Ida et al. used *o*-phthalaldehyde in the presence of 2-mercaptoethanol and sodium hypochlorite as a derivatization agent. This method was found to be suitable for evaluating the cysteamine and cystamine content in various organs and tissues [74].

#### 3.2.3. Enzymatic Essay

This assay consists of the inhibition of d-amino acid oxidase enzyme activity by the product of the reaction occurring between cysteamine and bromopyruvate. Thus, the enzyme activity is proportional to the amount of cysteamine [84]. In 1987, Duffel et al. described a new method based on the oxidation of cysteamine to hypotaurine by the action of cysteamine dioxygenase. Consequently, the quantification of oxygen uptake was proportional to cysteamine concentration [85].

#### 3.2.4. Gas Chromatography

The most used derivative agents for quantification of cysteamine by flame ionization are pivaldehyde (2,2-dimethylpropanal) and (trimethylsilyl) trifluoracetamide [65,66]. On the other hand, regarding flame photometric detection, isobutylchloroformate was used as the derivatization agent in a sensitive and selective method developed by Kataoka et al. [73,75]. Derivation agents, carrier gas, columns and temperature conditions of gas chromatography methods are described in Table 4.

#### 3.2.5. Electrochemical Detection

As previously commented, cysteamine is easily oxidized to cystamine, a process that can be electrochemically detected. These types of reactions consist of the gain or loss of electrons followed by subsequent rearrangements. Diverse electrodes have been used. Kelly et al. [86] described the first electrochemical detection of cysteamine when cysteamine was first analyzed by HPLC with an electrochemical detector with a platinum electrode or a single gold/mercury electrode [87,88]. Later, modified electrodes have been used due to high overpotential and low electrical signals associated with the use of unmodified electrodes. Raoof et al. fabricated a functionalized single-wall carbon nanotube modified glassy carbon electrode [89]. In later years, several research groups worked on the development of new modified electrodes: carbon paste electrode [90,91,92,93,94,95], multiwall carbon nanotubes paste electrode [96,97,98,99] and screen-printed electrode [100]. The modified electrodes mentioned above are summarized in Table 5.

Electrochemical methods showed a linear relationship between the catalytic oxidation peak and cysteamine concentration, being applicable in certain concentration ranges and different samples [60].

## 4. Current Situation and Future Perspectives

At present, and as previously mentioned, only two presentations are available on the market for the treatment of ocular cystinosis despite new developments in the field of cysteamine DDSs. The first commercialized one was Cystaran^®^, which consists of an aqueous solution of 0.44% cysteamine, and the second, Cystadrops^®^, whose composition is sodium carmellose, which provides greater viscosity and biopermanence on the ocular surface. Due to this addition, Cystadrops^®^ presents a more comfortable dosage schedule (administrated every 4 h) than Cystaran^®^ (administered every hour while awake) [21,22,23]. Along with these presentations, in the last few years topical compounding formulations prepared by hospital pharmacy services have also been used for the treatment of ocular cystinosis [24,44]. Many of these compounding formulations lacked clinical trials to evaluate their efficacy in clinical practice.

Nowadays, at clinicaltrials.gov there is only one ongoing clinical trial recruiting for the evaluation of a new therapy for the treatment of cystinosis. This study is a phase 1/2 clinical trial (NCT03897361) that will assess the safety and efficacy of enriched gene-corrected hematopoietic stem cells isolated from patients affected with cystinosis. The subjects receive marrow cytoreduction with busulfan prior to infusion of the genetic investigational product CTNS-RD-04 [101].

According to a quality assessment of investigational studies in the therapeutic use of cysteamine in cystinosis, only a few references provide an adequate quality level in order to stablish an acceptable strength of recommendation [102]. At present there is only one randomized clinical trial that could report on the efficacy and safety of cysteamine eye drops in preventing eye complications associated with cystinosis [20]. However, this trial does not have a sufficient quality level to issue a recommendation about cysteamine use in this indication. All other studies [103,104,105,106], although classified as randomized studies, cannot be considered as such, as they do not employ a real randomization per patient. These studies employ a per eye randomization so that the same patient receives the intervention treatment in one eye, and the control in the other. This, while a common practice in ophthalmology clinical trials, is not recommended [107], as it does not imply a real randomization as to the effect that this element of clinical research study design can have on bias reduction.

The only randomized clinical trial available [20] compared a viscous commercial presentation with an hydrochloride cysteamine concentration of 0.55% with a hospital pharmacy formulation of 0.10%. It used as its primary end point the difference from the baseline density assessment of cysteamine crystals measured through IVCM. Photophobia was included as a secondary end point. In the results analysis, taking into account the number of eyes rather than the number of patients, there was a loss of assessment of 33% of the eyes in the intervention group and 47% of the eye evaluations of the control group. In this analysis, the preparation in the form of 0.55% viscous gel found a 40.4% reduction in density, compared to 0.7% in the control group. The selected primary variable was a subjective variable, which would have been required to improve the internal validity of the analysis as a blind study. Instead, the design of the study was open, which increased uncertainty about the results obtained. In addition, the loss of patients could have resulted in the invalidity of the results, while the selection as a comparator of a concentration formulation more than five times lower could have justified the differences.

A recent systematic review and meta-analysis concluded that the viscous formulation of cysteamine 0.55% was better than standard formulation of cysteamine 0.1% in terms of efficacy. The lack of efficacy of the standard formulation might have been due to use of low dose (0.1% in the standard formulation group compared to 0.55% in the viscous formulation group). However, relative stability and requirement of less frequent application of the viscous solution gave an added advantage. Although, occurrences of local adverse events were more in the viscous formulation group, they were manageable. More trials with equivalent doses between the standard and viscous formulation to evaluate the comparative safety and efficacy are needed [16,107]. However, despite the low quality evidence, prospective observational studies suggest that cysteamine continues to be considered as a useful therapeutic strategy to delay ocular signs and symptoms associated with this systemic pathology [1].

At present, in order to overcome the main drawbacks linked to the use of conventional eye dosage forms, new DDSs are being developed for the treatment of ocular cystinosis. In spite of the fact that significant advances have been made in the area, there are still some unexplored alternatives which must be studied because of their promising in vivo results in other diseases, such as liposomes and nanoparticles developed for the treatment of glaucoma or age-related macular degeneration [36,38,108].

Nowadays, other pathways have been recently explored to improve the treatment of ocular cystinosis. Andrzejewska et al. hypothesized that cystinosin could have other roles in addition to transporting cystine out of the lysosome. In this sense, they found that the mammalian target of the rapamycin complex 1 (mTORC1) pathway, which serves as a core regulator of cellular metabolism, proliferation, survival and autophagy, was downregulated in proximal tubular cell lines derived from Ctns2/2 mice. Decrease of lysosomal cystine levels due to the administration of cysteamine did not rescue mTORC1 activation in these cells, suggesting that the downregulation of mTORC1 was due to the absence of cystinosin rather than to the accumulation of cystine. In conclusion, their results showed a dual role for cystinosin as a cystine transporter and as a component of the mTORC1 pathway, and highlighted the need to develop new treatments not only dependent on lysosomal cystine depletion [109].

Another approach to the improvement of ocular cystinosis treatment was carried out by Thoene et al. The therapeutic use of transmembrane proteins, such as cystinosin, is limited because of irreversible denaturation when they are not in their native lipid membrane. In this regard, these authors showed that microvesicles containing functional cystinosin were spontaneously produced by infecting Spodoptera frugiperda cells (Sf9) with baculovirus containing human wild-type CTNS. Furthermore, they demonstrated that the addition of such microvesicles containing cystinosin green fluorescent protein (GFP) to ex vivo rabbit ocular globes yielded punctate perinuclear green fluorescence in the corneal keratocytes. These results support potential therapeutic use of these cystinosin containing microvesicles in treating ocular cystinosis with the advantage of administering twice per month instead of daily topical administration. However, further preclinical and clinical trials should be done in order to confirm the obtained results [110].

Finally, in 2020 Hollywood et al. generated the first human induced pluripotent stem cell (iPSC) and kidney organoid models of cystinosis. They used several techniques to examine hallmarks of cystinosis, including cystine accumulation, lysosome size, the autophagy pathway and apoptosis in the cystinosis models. Compared with controls, these cystinosis models exhibited elevated cystine levels, increased apoptosis and defective basal autophagy. On the one hand, cysteamine treatment ameliorated this phenotype except for abnormalities in apoptosis and basal autophagy. On the other hand, they found that treatment with everolimus, an inhibitor of the mTOR pathway, reduced the number of large lysosomes, decreased apoptosis and activated autophagy, but it did not rescue the defect in cystine loading. Accordingly, dual treatment of cystinotic iPSCs or kidney organoids with cysteamine and everolimus corrected all of the observed phenotypic abnormalities. Consequently, the authors suggested that combination therapy with cysteamine and everolimus may be the future of cystinosis treatment, improving upon current therapeutic treatments [111].

## 5. Conclusions

Cystinosis is a rare metabolic disease for which, even today, there is no definitive cure. To prevent its progression, cysteamine is systemically administered allowing the formation of the cysteamine-cysteine complex, which prevents the accumulation of cystine crystals in the body’s tissues. To achieve effective concentrations in the cornea, its topical administration is required. Two presentations are marketed as ophthalmic drops (Cystaran^®^ and Cystadrops^®^), which present complicated dosage schedules that make treatment compliance difficult. For all these reasons, it is important to develop new DDSs that would allow increasing the corneal residence time, ocular bioavailability and comfort for the patient. This development remains challenging, since cysteamine is a very sensitive compound to oxidation, and whose analytical determination is complex.

In the last decades, different research groups have focused on the development of hydrogels, nanowafers and contact lenses, which allow a sustained release of cysteamine for longer periods of time compared to eye drops. However, new DDSs, which have offered promising possibilities in other diseases such as liposomes and nanoparticles, must be explored. For the characterization of ophthalmic formulations, it is important to have a good analytical determination method, although most of the proposed ones have not been developed for this objective. Even so, more studies are still needed to achieve optimal cysteamine release that favors comfortable dosage schedules without compromising drug stability, and thus offer a definitive ophthalmic treatment that prevents the advance of the ocular cystinosis, as well as exploring new therapeutic alternatives that allow the disease to be addressed by other pathways.

## Figures and Tables

**Figure 1 pharmaceutics-12-01177-f001:**
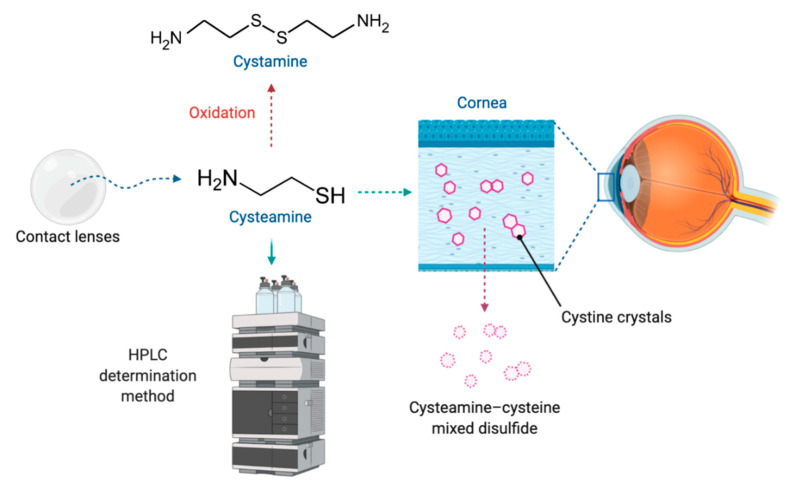
Cysteamine released from a drug delivery system to the ocular surface, where it forms the complex cysteamine-cysteine, facilitating the removal of cystine crystals from the cornea. Cysteamine is easily oxidized to cystamine, and can be detected from several determination methods, such as high-performance liquid chromatography (HPLC).

**Figure 2 pharmaceutics-12-01177-f002:**
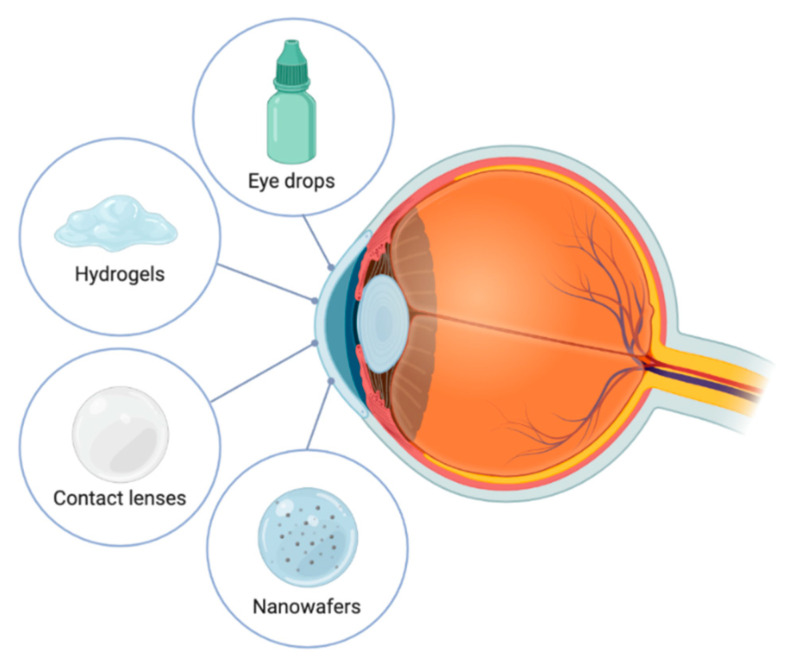
Drug delivery systems developed for cysteamine ophthalmic administration.

**Table 1 pharmaceutics-12-01177-t001:** Hydrogels developed as cysteamine delivery systems.

Polymer Name	Polymer Type	% Released Cysteamine	Time	Reference
Carbomer 934	Synthetic	80	210 min ^1^	[5]
Hyaluronic acid	Natural	60.7	24 h	[9]
88% Deacylated gellan gum and 12% kappa carrageenan	Natural	36.3	24 h	[9]
Carbomer 934	Synthetic	80	20 min	[10]
Hydroxyethyl cellulose	Natural	80	14 min	[10]
Hyaluronic acid	Natural	80	14 min	[10]
Hydroxypropylmethyl-cellulose	Natural	81.2	8 h	[33]

^1^ Cystamine–phenylalanine conjugate.

**Table 2 pharmaceutics-12-01177-t002:** Contact lenses developed as cysteamine delivery systems.

Commercial Name	Material	Diffusion Barrier	Total Cysteamine Release Amount (μg)	Release Duration	Reference
1-DAY ACUVUE^®^ TruEye™	Narafilcon B (silicone hydrogel)	10.22% VE ^1^	600.1 ± 27.2	25 min	[50]
1-DAY ACUVUE^®^ TruEye™	Narafilcon B (silicone hydrogel)	22.24% VE	527.7 ± 21.7	90 min	[50]
ACUVUE^®^ OASYS^®^	Senofilcon A (silicone hydrogel)	19.14% VE	408.8 ± 33.9	3 h	[50]
1-DAY ACUVUE^®^ TruEye™	Narafilcon A (silicone hydrogel)	10% VE	651 ± 31.2	0.90 h	[52]
1-DAY ACUVUE^®^ TruEye™	Narafilcon A (silicone hydrogel)	20% VE	603.1 ± 21.2	2 h	[52]
1-DAY ACUVUE^®^ TruEye™	Narafilcon A (silicone hydrogel)	30% VE	538.9 ± 17.3	4 h	[52]
ACUVUE^®^ OASYS^®^	Senofilcon A (silicone hydrogel)	10% VE	464.0 ± 8.6	0.89 h	[52]
ACUVUE^®^ OASYS^®^	Senofilcon A (silicone hydrogel)	20% VE	408.1 ± 36.7	2.15	[52]
ACUVUE^®^ OASYS^®^	Senofilcon A (silicone hydrogel)	30% VE	345.0 ± 33.2	4.25 h	[52]
Noncommercial	-	0.3% CB ^2^	148 ± 10	10 min	[56]
Noncommercial	-	0.3% CB + 20% VE	123 ± 7	40 min	[56]

^1^ VE: vitamin E; ^2^ CB: carbon black.

**Table 3 pharmaceutics-12-01177-t003:** HPLC detection methods for cysteamine.

Derivatization Agent	Stationary Phase	Mobile Phase	Flow Rate (mL/min)	Detector ^1^	T (°C)	Elution Time (min)	Limit of Detection	Ref.
2-chloro-1-methylquinolinium tetrafluoroborate	C_18_ (5 μm; 4.6 mm × 150 mm)	Gradient elution or isocratic elution (trichloro acetic acid and acetonitrile)	1	UV	25	9	0.1 μM	[76]
Monobromo-bimane	C_18_ (5 μm; 4.6 mm × 150 mm)	Gradient elution (methanol, acetic acid and water)	1.5	FL	RT	12.5	nmol	[68]
Monobromo-bimane	C_18_ (3 μm; 4.6 mm × 150 mm)	Acetonitrile	1.5	FL	RT	4.3	50 nM	[69]
Monobromo-bimane	C_18_ (5 μm; 2.1 mm × 100 mm)	Water: methanol (65:35)	0.3	FL	-	11	2 nM	[70]
Ammonium 7-fluorobenzo-2-oxa-1,3-diazole-4-sulphonate	C_18_ (8–10 μm; 3.9 mm × 300 mm)	Gradient elution (methanol and sodium acetate)	1	FL	RT	10	0.07 pmol	[71]
Ammonium 7-fluorobenzo-2-oxa-1,3-diazole-4-sulphonate	C_18_ (5 μm; 2.0 mm × 250 mm)	Phosphate buffer: CH_3_CN (96:4)	0.3	FL	30	5	0.47 μM	[72]
*N*-(1-pyrenyl) maleimide	C_18_ (5 μm; 4.6 mm × 250 mm)	Acetonitrile: water (70:30)	1	FL	RT	10	0.01 nM	[77]
7-chloro-*N*-[2-(dimethylamino)ethyl]-2,1,3- benzoxadiazole-4-sulfonamide	C_18_ (2 nm, 4.6 mm × 150 mm)	Gradient elution (water, acetonitrile and trifluoroacetic acid)	0.6	FL	50	6.4	154 fmol	[79]
4-fluoro-7-sulfamoyl benzofurazan	C_18_ (3 μm; 3.9 mm × 150 mm)	2.5% methanol and ammonium acetate	1	FL	-	-	-	[80]
6-aminoquinolyl-*N*-hydroxysuccinimidyl carbamate	C_18_ (5 μm; 2.1 mm × 150 mm)	Gradient elution (sodium acetate and trimethyl- amine, acetonitrile and water)	0.3	FL	RT	29	0.77 pmol	[81]
-	C_18_ (5 μm; 4.6 mm × 250 mm)	NaHpSO in phosphoric acid:acetonitrile (85:15)	1	UV	25	6.9	0.032 μg	[64]
5,5′-dithiobis-(2-nitrobenzoic) acid	C_18_ (5 μm, 4.6 mm × 250 mm)	Gradient elution (formic acid and acetonitrile)	1.0	UV	RT	26	3.3 mg	[82]

^1^ FL: HPLC coupled with fluorescence detection; UV: HPLC coupled with UV detection; RT: Room temperature.

**Table 4 pharmaceutics-12-01177-t004:** Gas chromatography detection of cysteamine.

Derivatization Agent	Carrier Gas	Column Used	Temperature (°C)	Limit of Detection	References
Pivaldehyde ^1^	Helium (35 mL/min)	5% SE-30 5′ × 1/8”	From 80° to 250 °C at 10°/min	8 pmole	[65]
(Trimethylsilyl) trifluoro acetamide^1^	Helium (80 mL/min)	2% SE-30 6 ft. ¼	From 75° to 230 °C at 8°/min	Sub nanomole	[66]
Isobutyl Chloroformate ^2^	Nitrogen (8 mL/min)	DB-210 15 m × 0.53 mm	From 170° to 250 °C at 5°/min	2 pmole	[73]
Isobutyl Chloroformate ^2^	Nitrogen (10 mL/min)	DB-210 15 m × 0.53 mm	From 170° to 250 °C at 5°/min	2 pmole	[75]

^1^ Flame ionization; ^2^ Flame photometric detection.

**Table 5 pharmaceutics-12-01177-t005:** Electrochemical detection methods.

Electrode	Mediator	Concentrations Range (μM)	Limit of Detection (μM)	References
Single-wall carbon nanotube modified glassy carbon electrode	1,2-*N*-aphthoquinone-4-sulfonic acid sodium	5.0–270	3.0	[89]
Carbon paste electrode	*N*,*N*-dimethylaniline/ferrocyanide	80–1140	79.7	[90]
Carbon paste electrode	(9, 10-dihydro-9, 10-ethanoanthracene-11, 12-dicarboximido)-4-Ethylbenzene-1, 2-diol and nickel-oxidecarbon nanotube	0.01–250	0.007	[91]
Carbon paste electrode	Ferrocene carboxaldehyde and nickel-oxide nanoparticle	0.09–300	0.06	[92]
Carbon paste electrode	Acetylferrocene and Nickel-oxide-carbon nanotube	0.1–600	0.07	[93]
Carbon paste electrode	*N*-(4-hydroxyphenyl)-3,5-Dinitrobenzamide and magnesium oxide nanoparticles	0.03–600	0.009	[94]
Carbon paste electrode	NiO dope Pt nanostructure hybrid (NiO–Pt–H)	0.003–200	0,0005	[95]
Multiwall carbon nanotubes paste electrode	Isoproterenol	0.3–450.0	0.09	[96]
Multiwall carbon nanotubes paste Electrode	Ferrocene	0.7–200	0.3	[97]
Multiwall carbon nanotubes paste electrode	3,4-Dihydroxycinnamic acid	0.25–400	0.09	[98]
Multiwall carbon nanotubes paste electrode	Promazine hydrochloride	Two dynamic ranges of 2.0–346.5 μM and 346.5–1912.5 μM	0.8	[99]
Screen printed electrode	La_2_O_3_/Co_3_O_4_	1.0–700.0	0.3	[100]

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
