# Peer review of "Recent Research in Ocular Cystinosis: Drug Delivery Systems, Cysteamine Detection Methods and Future Perspectives"

_pharmaceutics, 2020, doi:10.3390/pharmaceutics12121177_

Round 1

Reviewer 1 Report

This is a very good written review article that is well organized. It deals with an un-tackled subject which is the cystinosis.

I recommend its publishing after performing the following minor changes:

1- The title needs to be changes to reflect the drug delivery aspects in the review.

2- Please stress more and highlight the limitations of the conventional used eye dosage forms and please mention in the future perspectives the need for the use of novel delivery systems and formulations such as the liposomes and/or gelatin nanoparticles.

The authors can refer to:

ACS Omega. 2019 Dec 16;4(26):21909-21913. 

Int J Pharm. 2018 Jul 10;545(1-2):229-239.

3- In figure 1. correct the word "oxydation"

4- In line 400: Please correct the phrase " every working hour" . 

5- Examples of new drug delivery systems should be suggested in the conclusion as well.

6- The authors should also mention something like that the ethical issues sometimes should outweigh the economical and feasibility issues in case of the orphan drugs and rare diseases. 

Author Response

This is a very good written review article that is well organized. Ideals with an un-tackled subject which is the cystinosis. I recommend its publishing after performing the following minor changes:

Thank you very much for taking your time in reviewing our manuscript. We have written our responses after each of your comments.

  1. The title needs to be changes to reflect the drug delivery aspects in the review.
  • We completely agree with your consideration, we have changed the title to “Recent research in ocular cystinosis: drug delivery systems, cysteamine detection methods and future perspectives”.

  1. Please stress more and highlight the limitations of the conventional used eye dosage forms and please mention in the future perspectives the need for the use of novel delivery systems and formulations such as the liposomes and/or gelatin nanoparticles. The authors can refer to: ACS Omega. 2019 Dec 16;4(26):21909-21913 and Int J Pharm. 2018 Jul 10;545(1-2):229-239.

In order to highlight the limitations of the conventional used eye drops, we have added the following paragraph:

“Ocular drug delivery has always been a challenge due to the limitations of conventional eyedrops. The high tear turnover rate (1 µl/ml), loss of drug due to rapid blinking, reflex tear production, the tear-film barrier and the high nasolacrimal drainage are factors that limit the absorption of topically applied ophthalmic formulations. To increase the drug bioavailability, a higher residence time on the ocular surface needs to be achieved. During the last decades, research has been done on the development of DDSs offering longer retention and a sustained release of the drug molecule to pass through these barriers [29]”.

Reference number 29 was also added: Gote, V.; Sikder, S.; Sicotte, J.; Pal, D. Ocular Drug Delivery: Present Innovations and Future Challenges. J. Pharmacol. Exp. Ther. 2019, 370, 602–624, doi:10.1124/jpet.119.256933.

In addition, in the future perspectives section we have mentioned the need for the use of novel delivery systems and formulations such as the liposomes and/or gelatin nanoparticles by adding the following paragraph and your proposed two references:

“At present, in order to overcome the main drawbacks linked to the use of conventional eye dosage forms, new DDSs are being developed for the treatment of ocular cystinosis. In spite of the fact that significant advances have been made in the area, there are still some unexplored alternatives which must be studied because of their promising in vivo results in other diseases, such as liposomes and nanoparticles developed for the treatment of glaucoma or age-related macular degeneration [36,38,109]”.

  1. In figure 1. correct the word "oxydation"

The typo was corrected to “oxidation”.

  1. In line 400: Please correct the phrase " every working hour".

The sentence was modified to “every hour while awake”.

  1. Examples of new drug delivery systems should be suggested in the conclusion as well.

The next sentence has been added in the conclusion: “However, new DDSs which have offer promising possibilities in other diseases, such as liposomes and nanoparticles, must be explored”.

  1. The authors should also mention something like that the ethical issues sometimes should outweigh the economical and feasibility issues in case of the orphan drugs and rare diseases.

We appreciate your suggestion and we have added in the introduction the following paragraph: “Therapeutic options marketed for ophthalmic treatment are scarce. Although ethical issues sometimes should outweigh the economical and feasibility issues, the pharmaceutical industry does not allocate sufficient resources for the study of rare diseases and the development of orphan drugs. Because the commercialized cysteamine presentations are not available in most countries, hospital pharmacy departments are responsible for preparing “homemade” eyedrops as a therapeutic alternative [24]. The problem is that on many occasions, these formulations lack exhaustive stability controls under different storage conditions, being cysteamine a very easily oxidizable molecule [25]”.                                                                                                                                       

Reference number 25 was also added: Reda, A.; Van Schepdael, A.; Adams, E.; Paul, P.; Devolder, D.; Elmonem, M.A.; Veys, K.; Casteels, I.; van den Heuvel, L.; Levtchenko, E. Effect of Storage Conditions on Stability of Ophthalmological Compounded Cysteamine Eye Drops. JIMD Rep. 2017, 42, 47–51, doi:10.1007/8904_2017_77

Reviewer 2 Report

The review manuscript describes about current challenges in ocular cystinosis research and treatment. The manuscript is well structured and all-important aspects have been described in detail. However, there are some minor flaws which should be addressed before proceeding for the publication.

Decision: Major revision

Comments:

  1. The title of the manuscript is “Current challenges in ocular cystinosis research and treatment”, which seems manuscript will describe about challenges however the manuscript describes about recent research, drug delivery systems and treatment more than the challenges. Authors should rewrite an appropriate title to justify the topic which is written.
  2. Authors should also include details of ongoing clinical trial if any as a separate section.
  3. While discussing the DDS such as nanowafers, authors may describe about thermoreversible gel which is a potential ocular therapeutic delivery system for sustained release if applicable to cystinosis as well. Following is the reference which authors may cite;

Bhatt P, Narvekar P, Lalani R, Chougule MB, Pathak Y, Sutariya V. An in vitro Assessment of Thermo-Reversible Gel Formulation Containing Sunitinib Nanoparticles for Neovascular Age-Related Macular Degeneration. AAPS PharmSciTech, Aug 9,2019;20(7):281.

  1. Authors should also describe challenges in detail including challenges of individual DDS described in the manuscript.

Author Response

The review manuscript describes about current challenges in ocular cystinosis research and treatment. The manuscript is well-structured and all-important aspects have been described in detail. However, there are some minor flaws which should be addressed before proceeding for the publication.

Thank you very much for your considerations, we really appreciate them. We have written our responses after each of your comments.

  1. The title of the manuscript is “Current challenges in ocular cystinosis research and treatment”, which seems manuscript will describe about challenges however the manuscript describes about recent research, drug delivery systems and treatment more than the challenges. Authors should rewrite an appropriate title to justify the topic which is written.

We completely agree, we have changed the title to “Recent research in ocular cystinosis: drug delivery systems, cysteamine detection methods and future perspectives”.

  1. Authors should also include details of ongoing clinical trial if any as a separate section.

Thank you for your comment. In order to include this information, we have added the following paragraph in section “Current situations and future perspectives”:

“Nowadays, at clinicaltrials.gov there is only one ongoing clinical trial recruiting for the evaluation of a new therapy for the treatment of cystinosis. This study is a phase 1/2 clinical trial (NCT03897361) that will assess the safety and efficacy of enriched gene-corrected hematopoietic stem cells isolated from patients affected with cystinosis. The subjects receive marrow cytoreduction with busulfan prior to infusion of the genetic investigational product CTNS-RD-04”[102].

  1. While discussing the DDS such as nanowafers, authors may describe about thermoreversible gel which is a potential ocular therapeutic delivery system for sustained release if applicable to cystinosis as well. Following is the reference which authors may cite;

Bhatt P, Narvekar P, Lalani R, Chougule MB, Pathak Y, SutariyaV. An in vitro Assessment of Thermo-Reversible Gel Formulation Containing Sunitinib Nanoparticles for Neovascular Age-Related Macular Degeneration. AAPS PharmSciTech, Aug9,2019;20(7):281.

We appreciate your comment and we have added the following new paragraph in “Nanowafers” section, adding also your proposed reference:

“The development of this nanowafer means a very innovative and interesting alternative to properly control cysteamine release on the ocular surface. In this sense, other studies are needed above all exploring the possibilities that nanosystems, which have demonstrated successful results for other pathologies, could offer for the treatment of ocular cystinosis [36,38]”.

  1. Authors should also describe challenges in detail including challenges of individual DDS described in the manuscript.

In response to your suggestion we have added challenges of each individual DDS described:

  • Hydrogels:

“In spite of the fact that important advances have been made regarding the development of new hydrogels, further studies should be carried out. Specifically, regarding the development of smart hydrogels which can be very advantageous because they dramatically change their volume and other properties in response to environmental stimuli such as temperature or pH [35]. In addition, the synthesis of hydrogels based on nanocarriers could be a very promising strategy which has provided interesting results in other pathologies [36]”.

  • Nanowafers:

“The development of this nanowafer means a very innovative and interesting alternative to properly control cysteamine release on the ocular surface. In this sense, other studies are needed above all exploring the possibilities that nanosystems, which have demonstrated successful results for other pathologies, could offer for the treatment of ocular cystinosis [36,38]”.

  • Contact lenses:

“Despite the advances made in the development of CLs as cysteamine delivery systems, more studies are needed to allow their optimization. For this, it is important to select those CLs that have a more suitable composition for correct loading and release, and to develop methods that allow better control of release, without affecting any of the CL parameters and preserving the stability of the cysteamine”.

Reviewer 3 Report

In this review, the authors consider a high-impact topic in ocular research. In fact, the treatment of ocular cystinosis represents a therapeutic challenge to be addressed with advanced researches and in many countries the therapy is still based on galenic recipes prepared by pharmacists in hospitals.

The authors have collected the most important bibliographic citations on this topic. The description of the methods and results is however very concise without sufficient in-depth analysis. Furthermore, the  more detailed  part is the analytical one on the active ingredient which was included in the review as a chapter completely separate from the scientific context of the selected literature on cystinosis.

The two section should be better integrated as the title of the review;  "Current challenges in ocular cystinosis research and treatment "does not refer to such an important analytical part.

In my opinion the authors should largely revise the structure of the paper and add more description on the different experimental methods used by the different authors to demonstrate the quality of their research.

Author Response

Thank you very much for your suggestions. In order to better integrate the different sections in the title of our manuscript, we have changed it to: “Recent research in ocular cystinosis: drug delivery systems, cysteamine detection methods and future perspectives”.

We thoroughly researched all the available information about cysteamine detection methods in order to provide a comprehensive overview of all the existing possibilities and its major characteristics, including all the references in case more detailed information is needed. We really appreciate your commentary and we completely agree with you because this is a very interesting section which could be much more extensive. However, we think that if we describe every single method in detail, we would make this section very long comparing to the other parts of the manuscript making it the most important section, which is not our intention.

Round 2

Reviewer 2 Report

Authors have addressed comments successfully and I recommend this manuscript to be published.

Reviewer 3 Report

The authors replied to all the submitted questions appropriately.

The work is interesting, well detailed in all the different specific parts. The authors resumed the mains results  in the field of the cystinosis  reported by  many authors  pointing correctly the  chemical aspects.